# How to Make the Unpredictable Foreseeable? Effective Forms of Assistance for Children with Autism Spectrum Disorder (ASD) during the COVID-19 Pandemic

**DOI:** 10.3390/diagnostics13030407

**Published:** 2023-01-22

**Authors:** Jagoda Grzejszczak, Agata Gabryelska, Agnieszka Gmitrowicz, Dominik Strzelecki

**Affiliations:** 1Department of Child and Adolescent Psychiatry, Medical University of Lodz, 92-216 Lodz, Poland; 2Department of Sleep Medicine and Metabolic Disorders, Medical University of Lodz, 92-215 Lodz, Poland; 3Department of Affective and Psychotic Disorders, Medical University of Lodz, 92-216 Lodz, Poland

**Keywords:** ASD, children and adolescents, COVID-19, telemedicine

## Abstract

Symptomatology in patients with the diagnosis of autism spectrum disorder (ASD) is very heterogeneous. The symptoms they present include communication difficulties, behavior problems, upbringing problems from their parents, and comorbidities (e.g., epilepsy, intellectual disability). A predictable and stable environment and the continuity of therapeutic interactions are crucial in this population. The COVID-19 pandemic has created much concern, and the need for home isolation to limit the spread of the virus has disrupted the functioning routine of children/adolescents with ASD. Are there effective diagnostic and therapeutic alternatives to limit the consequences of disturbing the daily routine of young patients during the unpredictable times of the pandemic? Modern technology and telemedicine have come to the rescue. This narrative review aims to present a change in the impact profile in the era of isolation and assess the directions of changes that specialists may choose when dealing with patients with ASD.

## 1. Autism Spectrum Disorder (ASD)—Multicausality, Heterogeneous Picture, Long-Term Consequences

ASD is a concept defined as a constellation of deficits that appear in early life, involving social communication and repetitive patterns of sensory-motor behavior [1]. It is estimated that 1 in 44 children currently meets the criteria for spectrum disorder, which may represent about 1% of the child population in Western societies [2,3]. More and more studies are clarifying the causes of the disorder. The estimated heritability of ASD is as high as 90% [3]. Up to 40% of children with a diagnosis of ASD have a diagnosis of a genetic syndrome or have chromosomal abnormalities [3]. A number of variants of genetic abnormalities make up the clinical picture of the disorders’ heterogeneity [4]. Confirmation of the important role of genetic factors in the development of ASD does not facilitate clinical management, and we currently do not have a dedicated algorithm for the therapeutic management of this disorder. The contribution of molecular genetics to the overall prognosis remains moderate due to the complex interaction between heredity and environmental factors [5]. Although therapeutic methods for people with ASD continue to develop, it is virtually impossible to eliminate all symptoms of the spectrum [6]. In particular, difficulties in social communication mean that a child or adolescent with a diagnosis of ASD will be a future adult with a strain at professional or relational levels. Mental health problems are also not excluded, although statistics on this problem are not precise [1,7].

## 2. Appropriate Diagnosis and Treatment

The diagnosis of ASD requires the involvement of a multidisciplinary team of experts [8]. It is established based on the diagnostic criteria for ASD contained, respectively, in the Diagnostic and Statistical Manual of Mental Disorders, 5th edition (DSM-5) for the United States and the International Statistical Classification of Diseases and Related Health Problems, 10th edition (ICD-10) for other countries. It should be mentioned that World Health Organization (WHO) member states, as of 1st of January 2022, can use the next, 11th version of the classification (ICD-11), in which the criteria for diagnosis, as well as the nomenclature of ASD itself, has changed noticeably [9,10]. In several countries, e.g., Poland, preparations to implement the ICD-11 criteria are still underway.

The latest versions of the DSM and ICD allow better specification of a patient’s difficulties with a diagnosis of ASD, which is inherently heterogeneous [9]. Both classifications mention the existence of four different profiles in the verbal part of ASD, describing them as combinations of either spared or impaired functional language and intellectual abilities [11]. Asperger Syndrome and Rett syndrome diagnoses are also disappearing from the nomenclature in ICD-11, which was already contained in the DSM in 2013 [12].

During the ASD diagnostic process, dedicated tools are used in addition to the knowledge and experience of clinicians. These include the Autism Diagnostic Interview-Revised (ADI-R), the Gilliam Autism Rating Scale—Third Edition (GARS-3), the Diagnostic Interview for Social and Communication Disorder (DISCO), the Developmental, Dimensional, and Diagnostic Interview (3di), the Autism Diagnostic Observation Schedule—Second Edition (ADOS-2), and the Childhood Autism Rating Scale—Second Edition (CARS-2) [13]. ADOS and its 2nd edition—a semi-structured, standardized assessment tool for people with suspected ASD—is considered part of the gold standard for diagnostic assessment [14]. It helps in diagnosing but also in assessments of the symptoms’ severity and the dynamics of responses to therapeutic interventions.

One of the critical clinical problems concerning ASD is the lack of dedicated treatment; then, symptomatic treatment is commonly employed. The importance of both pharmacological and non-pharmacological treatment should be emphasized [15]. Pharmacological treatments include psychostimulants, atypical antipsychotics, antidepressants, and alpha-2 adrenergic receptor agonists [15,16]. The use of medications provides an opportunity to quickly control acute symptoms, including aggressive or auto-aggressive behavior. Yet, non-pharmacological interactions are the key. Among the most common is cognitive-behavioral therapy (CBT) [15], which works especially well for patients with a pronounced anxiety component [17]. Among the well-known and widely used therapeutic methods with good results in patients with ASD, we can also include music therapy, animal therapy, or mind–body therapies [18,19,20]. Long-used, yet controversial, treatments include intervention methods informed by the principles of applied behavior analysis (ABA) [21]. The use of medical cannabis still seems contentious [22]. While there is evidence for the efficacy of pure cannabinoids (cannabidiol, CBD), commonly used remedies also contain Δ9-tetrahydrocannabinol (THC), which, especially in the child and adolescent population, may be associated with an "entourage effect", and thus, recreational use of cannabinoids might follow [23]. Also promising appears to be the administration of intranasal oxytocin, for which effectiveness in improving social functioning has been shown in recent studies [24]. According to studies, dietary supplements or nutritional interventions do not produce noticeable results [25]. The effectiveness of interventions in the microbiota still requires further research [26]. With the rapid development of clinical genetics and the emphasis on the genetic etiology of ASD, gene therapies seem promising [27]. Nevertheless, the polygenic inheritance of ASD may prove to be a problem [28].

The wide range of therapeutic methods reflects the diversity of the population of children with an ASD diagnosis and highlights the need for more structured algorithm treatment [29].

## 3. The Importance of Environmental Stability

The early diagnosis of ASD in children is crucial since the brain neuroplasticity during this developmental period allows effective interventions to alleviate the symptoms [30]. A child’s condition cannot be described in isolation from the family system [31]. Studies on a population of children without ASD suggest that stressful events in childhood can cause structural and biochemical changes in the brain, which consequently cause cognitive impairment in children and, further, in adults [32]. Children with neurodevelopmental disorders have been shown to be statistically more likely to have negative experiences in childhood [33]. It has also been shown in children with ASD that environmental factors, such as financial instability, can contribute to cognitive problems [33]. A stable relationship with a parent for any child is a protective factor [34]. Studies show that parents of children with ASD present higher levels of stress, exhibit lower parental efficacy, are more likely to face mental and physical health problems, and are more likely to divorce [35,36]. Deficits in social skills are one of the main features of ASD. At the same time, participation in the daily activities of the family system or the school community is fundamental to the child’s more appropriate development [37]. Expectations of parents with ASD children focus on hopes for independence, happiness, and improved skills for their children (language and speech abilities, dealing with their own emotions (e.g., anxiety, frustration), controlling difficult behaviors, incl. aggression), increased authentic, socially meaningful relationships, and future employment [38]. Studies show that reduced emotionality in the neurodevelopmental child population is associated with a lower risk of behavioral or emotional problems [39]. In turn, lower levels of social competence may generate anxiety disorders more frequently than in the general population [40]. Research has shown that parents’ attitudes toward the socialization of a child with ASD are not insignificant—the more parents are focused on correcting emotional experiences, the higher the child’s anxiety level [41]. Statistics also report higher levels of stress in parents with ASD themselves compared to other groups of parents [42]. U.S. researchers have also demonstrated the different nature of communication by parents of neurodevelopmental twins compared to parents of typically developing (TD) children. Caregivers of children with ASD produced less causal talk and proportionally less desire and cognitive interaction than caregivers of TD children [43].

## 4. Difficulties during the COVID-19 Pandemic

Previous research indicates that children and adolescents tend to have a mild COVID-19 course with a good prognosis [44]. However, it is important to lean into the indirect, long-term effects of the pandemic on the child and adolescent population. It has to be mentioned that the mental health and well-being risks, family income disruption, and attendant stressors, including increased family violence, delayed medical care, and the critical issue of prolonged loss of face-to-face learning in a regular school environment cause a number of serious health, social, and economic consequences [45]. Difficulties in social communication can simultaneously strain the parent–child relationship [31]. Difficulties and facilities for children/youth diagnosed with ASD during the COVID-19 pandemic are detailed in Table 1.

Epidemiological restrictions during the pandemic led to a disruption in daily routines and a reduction in the availability of facilities attended for treatment; children with ASD are particularly affected by these changes [46]. In addition, it is essential to remember the co-occurrence of psychopathological symptoms/syndromes in children and adolescents with a diagnosis of ASD. A study in the U.S. state of Maryland found that 59% of children with a neurodevelopmental issue experienced worsening of their pre-pandemic psychiatric diagnosis (irritability (28%), sleep problems (24%), anxiety (12%), and disruptive behavior (11%)) and developed new psychiatric symptoms during the pandemic (irritability (26%), anxiety (22%), and sleep problems (19%)) [47]. Brazilian researchers have attempted to summarize the impact of isolation during the pandemic on psychopathological symptoms in children/youth with ASD. The analysis of data from 8610 patients with ASD showed that behavioral disorders and disturbed sleep patterns increased during the pandemic [48]. Data from before the COVID-19 isolation period indicate that two-thirds of the pediatric population suffers from chronic insomnia [49]. The results collected by French researchers from 239 ASD patients under the age of 21 confirmed an increase in sleep problems and behavioral disorders, as well as provided information on the deterioration of social functioning in this pool of patients [50]. The statistics on the hospitalization parameters (admission risk, its prolonged duration) of young patients with a diagnosis of ASD with comorbid intellectual disabilities (ID) seem disturbing [51]. Qatari data collected from 58 caregivers of children with ASD during the social constraints of isolation in the pandemic era (reduced availability of therapeutic facilities, need for greater parental involvement) indicate that 41% of parents surveyed felt an increase in the burden of caring for their children. Interestingly, a slight decrease in aggression levels in ASD patients was also observed [52]. Statistics provided by the Italians also seem optimistic. No significant worsening in adaptive functioning or problematic and repetitive behaviors emerged after the compulsory home confinement. In the population of school-aged children, stability in clinical status was observed, while in the preschool group, improvement in adaptability was even noted. The main limitation of this data is the small study group, which included only 85 Italian ASD children [53].

Are adults prepared to help the younger generation with neurodevelopmental difficulties? 

## 5. Diagnostic and Therapeutic Innovations Useful during Isolation

Despite the social isolation, the intensification of parental interactions, the continuation of therapeutic interventions in the online form, as well as the crisis intervention itself available in the form of remote telemedicine—despite earlier availability—has now significantly accelerated [53]. The use of telemedicine in ASD diagnosis was estimated to increase from 6 to 78% [54]. Although, prior to the pandemic period, there were attempts to use telemedicine in the diagnosis of children with ASD, this was rather limited to laboratory settings. The pandemic is changing the approach and stimulating research into new tools. It turns out that it is possible to make a diagnosis even based on amateur videos recorded by parents [55]. The use of technology to make a diagnosis is possible, with a sensitivity of 96%, and it also allows for the immediate implementation of therapeutic interventions [55]. One of the new helpful tools is TELE-ASD-PEDS (TAP) [56]. TAP is an application designed for use by providers and families during a telehealth assessment for autism. Statistics were based on 197 ASD diagnostic teleassessments, demonstrating the diagnostic utility of this tool [54]. The usefulness of another diagnostic tool is currently being tested by the Italians. The Web Italian Network for Autism Spectrum Disorder (WIN4ASD) is the first Italian online web-based screening tool. A screening-scoring algorithm was integrated into the web platform, which can generate an automatic and immediate individual patient output that significantly speeds up the diagnosis. Currently, the application has over 450 registered pediatricians and is experiencing exponential growth [57]. Beginning and continuing therapeutic interventions (especially CBT therapy) has proven crucial. Anxiety disorders very often co-occur with a diagnosis of ASD. The introduction of the necessary modifications, in the form of conducting therapy (telemedicine) as well as the updating of the goals of treatment (pandemic as a new source of anxiety), has allowed for the continuation of therapeutic interactions in reality and adapted to isolation—without breaking the continuity and thus worsening of the mental state of children [58]. Clinical practice has shown that even judo lessons could be conducted remotely. This experiment was conducted with 9 high school students from the state of Florida in the US. All of them expressed satisfaction with the maintained routine as well as the enjoyment of the physical activity. The lack of space turned out to be the main disadvantage. However, the home isolation itself only required a change in the form of instruction, without having to change the operating pattern [59]. Low-income families were hopeful about receiving benefits through future online services [60]. Despite the many difficulties experienced by parents of children with ASD during home isolation, it can be concluded that the COVID-19 pandemic may be an opportunity to practice parenting skills, improve the quality of communication, or strengthen family ties [61].

## 6. Change the Form of Assistance during the COVID-19 Pandemic and Other Situations with Limitations on the Availability of Classical Medical Assistance

Statistics show that nearly half of the caregivers of children with ASD used telemedicine services during the COVID-19 pandemic for the first time [62]. Services provided via telehealth included diagnostic assessments, preference assessments, early intervention and follow-up, applied behavior analysis (ABA), functional assessment, functional communication training, and parent training (Figure 1) [63]. It is also important to remember the patients for whom pharmacotherapy is necessary—due to serious behavioral disturbances (e.g., irritability), including severe tantrums, aggression, and self-injury—and should be continued and supervised in a telemedicine setting to maintain the reduction in the symptoms presented [64].

Public awareness is increasing, so more and more parents are seeking help from diagnostic specialists. Isolation during the pandemic has forced the health sector to look for diagnostic alternatives. ASD diagnosis has also found its place in the online space. Studies conducted in the U.S. show that the change in the diagnostic format is accepted by both parents of children with neurodevelopmental problems as well as specialists leading the diagnosis process, but it might be less accepted by younger preschool children [65]. Regular therapeutic interactions are also necessary, as well as crisis intervention. An Italian team has undertaken the development of a model that allows both lines of action to be carried out in remote conditions. An attempt was made to transfer already known intervention models to remote needs. It was proven that this form of intervention not only helped to maintain the stable psychological state of children/young adults with ASD but also strengthened parents’ sense of competence and increased the relevance of their interventions [66]. Geographic or economic conditions are not insignificant. Based on the work of speech-language therapists (SLTs), researchers at the University of South Africa have attempted to evaluate the effectiveness of the continuation of therapeutic interventions for children/youth with ASD. It was shown that a form of remote therapy not only reduced the travel costs associated with getting to the site of therapy services but also increased the satisfaction of both caregivers and clinicians with the effects of the activities provided [67]. Above all, it allows for an increase in the reach of the services offered, which is not insignificant, especially in developing countries [65,67]. The active participation of parents in the therapeutic process is essential. An international team of researchers attempted to evaluate the effectiveness of behavioral interventions through video conferencing, during which parents were given guidance. They noted an increase in self-efficacy in all participants whose parents faithfully followed the therapists’ comments. Yet, these results should be carefully considered, as the study group was comprised of only 4 participants [68]. In the U.S., an online survey of nearly 2000 parents examined whether broad access to information about ASD symptoms increases parental awareness and thus speeds up diagnosis [69]. The key, therefore, is free access to information and the exchange between parents and specialists, which is definitely accelerated by online communication [70]. Long before the COVID-19 pandemic, Canadian researchers hypothesized the importance of online support groups for parents of children with ASD, showing significant levels of satisfaction with this form of support in these parent groups [71]. The need for online parental support during the COVID-19 pandemic is confirmed by reports from Saudi Arabia [72]. Colleagues from Italy who decided to examine the essence of multidisciplinary therapeutic interactions necessary in caring for children/adolescents with ASD, launched the “Sistema Unitario in una Plataforma Educativa e Riabilitativa” (SUPER) program to facilitate communication between parents, teachers, and therapists using an online platform. The initial usability studies of the platform, carried out with a group of 30 adults involved in helping children with ASD, are very promising. Assessments using the System Usability Scale (SUS) showed that both parents and therapists rated the app as “excellent”. Researchers emphasized the utility of the tool as a space for communication, pointing out specialized recommendations and mutual support for parents, as well as a platform for collecting patient data [73]. Not uncommon in the era of the pandemic was the phenomenon of placing the burden of both the education and therapy of children with ASD on their parents at home. It turns out that, with this level of burden, online consultation is not enough. In Iran, a two-month, online internet course was conducted for a group of more than 300 parents of children with ASD to provide regular guidance and support. A range of online materials were used, including videotapes as well as synchronous recordings with instant instruction. The survey showed that 93% of parents were satisfied with this form of training and support. The remainder pointed out flaws, i.e., technical problems, poor quality of the internet connection, or costs associated with the use of modern technology [74]. Similar results are provided by research from China, where an online program for parents of children with an ASD diagnosis, Parent Education and Training (PET), has been introduced. Feedback from 294 PET users shows that virtual sessions conducted by trainers through the platform provided parents with information on how to better intervene when difficult symptoms occur, but also provided information on how to nurture a support network, especially given the multicultural nature of the Chinese population [75].

The usefulness of technological solutions—live video consultations, video recordings, or phone calls—can greatly facilitate, in the future, both diagnosis and therapeutic interactions. This is not only beneficial in the case of the need for home isolation of families where there are children with neurodevelopmental disabilities but also for those for whom the availability of specialists is limited, for example, due to the place of residence (rural area) [76,77]. Nevertheless, it is crucial to conduct further research on the use of telemedicine for ASD patients in a larger study group and in different geographic areas. A study on a subgroup of one of the largest US cohorts of children with ASD (SPARK) provides information about the need to combine remote and stationary impacts in the event of another crisis similar to the COVID-19 pandemic. The researchers emphasize that remote actions turn out to be insufficiently effective, especially in the cases of families with young children, a significant intensification of ASD symptoms, and those of low economic status [60]. It is important to remember that there are still interventions that are impossible to perform remotely. These include, for example, crisis intervention during agitation or aggression/autoaggression episodes. It is not possible to administer emergency medication online or to use coercive measures in the case of an immediate threat to the health and life of a young patient [78]. The ethical context of the services provided in this way is also not insignificant (the limitations of non-verbal communication in the patient–doctor dyad and limited opportunities to build and strengthen the therapeutic relationship) [79].

## 7. Conclusions

The COVID-19 pandemic has brought many changes to the health sector. Although most often children and adolescents underwent coronavirus infection in a mild symptomatic form, the impact of the pandemic on the youngest population may still be discernible in long-term observations [80]. Children/adolescents with a diagnosis of ASD are a specific patient group requiring the participation of a multidisciplinary treatment team as well as parental involvement [81,82]. In a situation where the level of anxiety due to the uncertainty of the scale of the threat in society is increasing, and home isolation is the primary protective measure, the quality of assistance for patients with ASD cannot be reduced. Available studies indicate that psychopathological symptoms (anxiety, affective disorders, self-destructive behavior, and sleep disorders) are increasing in the youngest population with ASD [83]. It is crucial to continue therapeutic interventions and provide access to diagnostic services for all psychiatric patients—children and adolescents with a diagnosis of ASD, in particular [84]. In this era of pandemics, telemedicine comes to the rescue [85]. In addition, increasing the outreach to ASD patients as well as collecting records in online accessibility can greatly accelerate the building of a research base with which to develop better and more effective forms of assistance. However, one should remember to use adequate and proven research tools and control the selection of the research sample as well as its dynamics [86]. Previous studies show that it is an effective alternative not only in the era of pandemics; thanks to its advantages, it can also be successfully used after the epidemiological crisis [87]. There is a need to continue research into the use of telemedicine to better refine algorithms based on it that can be used in usual life as well as in future crisis situations with children and adolescents with neurodevelopmental issues [88].

## Figures and Tables

**Figure 1 diagnostics-13-00407-f001:**
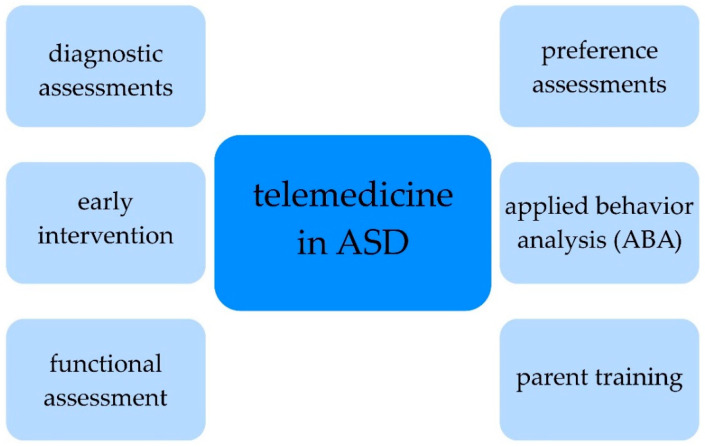
The use of telemedicine in children/adolescents diagnosed with ASD. Abbreviations: ASD, autism spectrum disorder; ABA, applied behavioral analysis. The figure shows the areas of use for telemedicine in children/adolescents with ASD and their caregivers, indicating its usefulness not only in the diagnostic or interventional processes but also in a wide therapeutic range or in assessing satisfaction with the provided hearing.

**Table 1 diagnostics-13-00407-t001:** Difficulties and facilities for children/adolescents diagnosed with ASD during the COVID-19 pandemic.

Difficulties	Facilities
disruption of the current plan of the day closure of inpatient therapeutic centers limited availability of diagnosticians limited availability of technology in less affluent families	limiting social contacts limiting social contacts’ access to therapeutic methods in a remote form—a greater range of services reduction of access to diagnostic /therapeutic centers among less affluent families a known constant environment of interaction easier communication with a multidisciplinary therapist team, parents

## Data Availability

Not applicable.

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
