# Peer review of "How to Make the Unpredictable Foreseeable? Effective Forms of Assistance for Children with Autism Spectrum Disorder (ASD) during the COVID-19 Pandemic"

_diagnostics, 2023, doi:10.3390/diagnostics13030407_

Round 1

Reviewer 1 Report

This is a review, so medical students, GPs, neurologist and other health professionals can benefit. 

Author Response

The answer is attached.

Reviewer 2 Report

The authors present a narrative review describing the effects of the COVID-19 pandemic on autistic individuals and their families. The authors support future development and continuation of Telehealth practices combined with in-person supports to support autistic individuals. The focus of the article seems to be on effective practices in the COVID-19 era, rather than a review of the literature on the specific impacts of COVID on autistic individuals.  Perhaps changing the title to better reflect the narrative review would be helpful for readers.  Overall, this review is a helpful summary of the needs of autistic individuals and the supports and services available during the COVID-19 pandemic and how these services may impact the future services delivery for individuals.

Recommend changing line 29 from "children currently suffers from a spectrum..." to "meets criteria for..." or other more supportive language.  Also the updated western prevalence is 1 in 44 rather than 1 in 59.  

For the paragraph describing assessments on line 61, these are not all the most up-to-date versions, please update.  For example, ADOS-generic is no longer used and GARS and CARS both are on the second version. 

Line 127- what is the purpose of this question? Recommend deleting.

Author Response

The answer is attached.
